# Exploring the Impact of Leader Bottom-Line Mentality on Subordinate Learning from Work Failures: A Social Information Processing Perspective

**DOI:** 10.3390/bs14030226

**Published:** 2024-03-11

**Authors:** Yunsheng Shi, Lei Gao, Haibo Yu, Shanghao Song

**Affiliations:** 1School of Government, Beijing Normal University, Beijing 100875, China; yunshengshi@mail.bnu.edu.cn (Y.S.); gaoleihr@mail.bnu.edu.cn (L.G.); yuhb@bnu.edu.cn (H.Y.); 2Curtin Business School, Curtin University, Perth 6102, Australia

**Keywords:** leader bottom-line mentality, psychological availability, learning from work failures, self-compassion, social information processing theory

## Abstract

Learning from work failures is not only beneficial for individual development but also crucial for improving organizational performance and achieving sustainable development. We hypothesize that leader bottom-line mentality, which is commonly used by leaders to prevent profit and performance losses, may reduce subordinates learning from work failures. Drawing on social information processing theory, this paper examines how and when leader bottom-line mentality negatively affects subordinates learning from work failures. We tested our hypotheses through a three-wave survey of 245 employees from several high-tech companies in China. For data analysis, we used SPSS 26.0 and Mplus 8.0 to test the theoretical model and research hypotheses. The results indicated that leader bottom-line mentality has a negative indirect effect on subordinates learning from work failures through the mediating role of subordinates’ psychological availability. In addition, subordinate self-compassion can mitigate this negative mediating mechanism. The present study has several theoretical and practical implications for the current literature.

## 1. Introduction

Failure is commonplace at work. Individuals can learn more from failures than from successes, with the insights gained from such setbacks contributing to greater success in future work endeavors [1]. Learning from work failures represents the specific processes that help individuals transform failures into learning experiences [2] and is an essential factor in transforming unsuccessful attempts into successes [3]. Learning from work failures is crucial not only for the improvement of personal work knowledge and skills [4] but also for the performance and development of organizations or teams [5]. Thus, the truest failure is not learning from work failures [6]. The existing literature has examined how individual characteristics such as personality traits [7,8], cognitive patterns [9,10], and emotional responses [11] affect individual learning from work failures. Few studies have focused on the impact of leader-related factors on employees learning from work failures. Leaders are known to play a central role in how employees cope with work failure and in organizational learning after failure [12,13]. Leaders can help employees effectively cope with and learn from work failure by providing a supportive and caring environment [13,14] or by using humor to alleviate the negative emotions associated with failure [15].

However, the existing literature mainly focuses on leader-related factors that may facilitate individual learning from work failures, while overlooking those that may hinder it. Due to the typical association of failure with performance declines and the resulting financial losses inflicted on organizations [16], the bottom-line mentality of leaders is prevalent as a means of ensuring organizational success [17]. Bottom-line mentality is described as “a one-dimensional thought pattern centered around ensuring bottom-line outcomes at the expense of competing priorities” [18]. Such a bottom-line mentality in leaders can undermine subordinates’ positive cognition about their work and foster negative emotions [19]. It can also undermine subordinates’ emotional commitment to their work, reduce their proactivity, and decrease their work engagement [20]. Leaders with a high bottom-line mentality, driven by a focus on high performance and profits, may neglect the learning opportunities provided by failure. Their emphasis on achieving success [20], coupled with increased monitoring and control of subordinates and promotion of competition among subordinates [21], may inhibit subordinates’ ability to learn from work failures. Consequently, in this study, we aim to examine the mechanisms by which leaders’ bottom-line mentality may reduce subordinates’ learning from work failures and to investigate the boundary conditions of this process.

Social information processing theory suggests that individuals’ attitudes and behaviors are largely influenced by their surrounding social context. They determine how to react by processing and interpreting particular social cues [22]. Leader bottom-line mentality signals to employees that leaders are more concerned with success, company profits, or financial objectives [23]. After observing or sensing signals of leader bottom-line mentality, subordinates may perceive greater risk in openly sharing work failures [2]; experience a more competitive atmosphere among colleagues [21], which is detrimental to constructing a psychological safety net in the face of failure [14]; and may exacerbate employee psychological exhaustion, leading to more negative emotions [19]. However, learning from work failures not only requires overcoming an individual’s aversion to failure but also poses a significant challenge to an employee’s self-esteem and self-confidence [7] and requires thorough preparation, a strong belief in oneself, and a positive affective state [6]. As a “sense of having the physical, emotional, or psychological resources to engage personally at a particular moment” [24], individuals with high psychological availability can increase their work engagement, performance, and creativity [25]. Therefore, we believe that a subordinate’s psychological availability plays an important mediating role between the leader bottom-line mentality and learning from work failures. Specifically, the signals transmitted through leader bottom-line mentality may decrease a subordinate’s psychological availability at work, thereby negatively affecting the efficiency of learning from work failures.

In addition, social information processing theory posits that people view their environment as a guide to their behavior and cognitions [22], and they will adjust their cognitions and behavior toward their work to match the environment [22]. Self-compassion, which is defined as “a self-attitude or a way people think of or relate to themselves” [26], is a productive way of dealing with distressing thoughts and emotions that produces mental and physical well-being [27]. Individuals with high levels of self-compassion are better able to positively manage the negative emotions caused by failure in the work process, mitigate the negative impact of failure events and leader bottom-line mentality, and allow their limited resources to be quickly invested in more meaningful things [28]. Leader bottom-line mentality is a typical contextual factor that sends signals to employees that they are overly focused on bottom-line goals and company profits, which further affects their attitudes and cognitions [18]. In this process, employees with different levels of self-compassion have different ways of coping with the environmental element (i.e., leader bottom-line mentality), and the resulting effects also vary significantly. Therefore, we argue that self-compassion may moderate the effect of leader bottom-line mentality on psychological availability.

In summary, based on social information processing theory, we constructed a moderated mediation model (see Figure 1) to examine how leaders’ bottom-line mentality affects subordinates’ learning from work failures and the mediating role of psychological availability as well as the boundary condition of self-compassion. Specifically, consistent with the empirical context of previous studies [15,29,30], and considering that the fierce competitive environment in China’s high-tech industry may cause managers to have a higher level of bottom-line mentality and that failure is more common in such industries, we used survey data from employees and their direct leaders in high-tech firms in China to test our hypothesized model. Through this research, we aim to make several theoretical contributions. First, we extend the applicability of social information processing theory to understand how leaders’ bottom-line mentality affects subordinates’ learning from work failures, thus responding to the call to explore “how leader-related factors affect employee learning from failure” [14]. Second, we discover the impact of leader bottom-line mentality on employees’ psychological states and behaviors in the context of failure, further enriching the literature on bottom-line mentality [20]. Finally, we examine key boundary factors that might mitigate the negative effects of leader bottom-line mentality, namely self-compassion. As a common human awareness [31], we enrich the application scenario of self-compassion in the field of organizational research.

## 2. Literature Review and Hypotheses

### 2.1. Learning from Work Failures

Failure is defined as “a deviation from goals, work standards, behavioral norms, or value judgments that can lead to a series of negative consequences” [5,32]. For individuals, failure is an inevitable experience. Individuals who experience failure need to invest more resources to solve the difficult problems inherent in failure, which helps to encourage individuals to proactively discover problems, learn knowledge, and master skills [4], thereby better coping with failure events and achieving positive work outcomes in the future. Therefore, compared to learning from successful experiences, individuals can learn richer content and gain deeper insights from failures [33,34]. From an individual perspective, scholars define learning from failure as “the specific process through which individuals not only detect or correct errors but also identify and explore their underlying causes” [35]. In view of the fact that learning from work failures has numerous positive effects on individual emotional experiences and work outcomes, scholars have focused on the antecedents that may influence employees’ learning from work failures.

In reviewing the existing research, most scholars have explored individual-level antecedents such as individual traits (e.g., narcissism; emotional stability) [7,8], cognitive patterns (e.g., failure attribution; regulatory focus) [9,10], and the emotional experience caused by failure (e.g., shame; anger) [11,15] and other individual factors that have an impact on learning from work failures. However, these studies ignore a key message: as a key figure in the functioning of a team or organization, the leader plays a critical and central role in how employees cope with failure [12,13]. From the perspective of social information processing theory, leaders can create an organizational or team learning atmosphere [36], which can convey positive work signals to employees, thereby facilitating subordinates’ learning [5]. Therefore, some scholars have paid attention to the impact of leader-related factors on subordinates’ learning from work failures and pointed out that in failure situations, leaders can tolerate employees [14], care for employees, and provide support to employees [13] or express humor to employees to resolve the negative emotions caused by failure [15], thereby helping employees effectively cope with failures and learn from them.

Unfortunately, these studies paid scant attention to why and how some leader-related factors may impede subordinates’ learning from work failures. As the most important line at the bottom of companies’ profit and loss number, leaders with a bottom-line mentality are usually extremely sensitive to profits and performance [23], and they will convey to subordinates “only focus on the bottom-line performance” signals; and will have a series of effects on employees’ psychological and behavioral responses [20]. Failure usually means performance decline or destruction and brings economic losses to the organization [16]. From this point of view, the occurrence of failure seems to go against the bottom-line mentality of leaders. As a result, the context of failure provides potential support for us to explore the impact of leader bottom-line mentality on subordinates learning from work failures.

### 2.2. Leader Bottom-Line Mentality and Subordinate Psychological Availability

Social information processing theory posits that individuals seek cues from their social environment, including their leaders, to ensure that their behavior is consistent with that environment [22]. At any given moment, psychological availability is disrupted by depletion of physical and emotional energy, individual insecurity, and outside life [24]. We propose that a leader’s bottom-line mentality will send negative information or cues to subordinates [23]. This information will exacerbate subordinates’ depletion of physical energy, increase emotional exhaustion [37], undermine their sense of individual job security [38], and thereby reduce their psychological availability.

First, a leader’s bottom-line mentality will lead to excessive depletion of subordinates’ physical energy. In the daily management process, a leader with a bottom-line mentality will often send a signal to subordinates that he/she is only concerned about profit and does not care about other things (e.g., subordinates’ health, personal development, and work-related happiness) [23]. Under the guidance of such information, when a subordinate fails at his job, the leader with a high bottom-line mentality will be concerned only with whether the bottom-line goals have been achieved or whether they are in jeopardy [39]. To ensure the achievement of these bottom-line goals, employees must devote more energy to work and meet performance goals by participating in mandatory overtime and high-intensity work [21]. These will undoubtedly increase the depletion of physical energy, reduce the ability of subordinates to pursue work happiness, and lead to health problems (e.g., insomnia) [21].

Second, a leader’s bottom-line mentality may be one of the signals that causes subordinates to have more negative emotions at work. Leaders with a high bottom-line mentality serve their self-interested pursuit of the performance bottom line in their management processes [23]. They signal to subordinates that ensuring the bottom line is paramount and tend to frequently monitor subordinates’ behavior to ensure that efforts are directed toward achieving the bottom line [18]. When faced with work failures, a leader with a high bottom-line mentality, in an effort to protect the bottom line, imposes greater work and performance pressure on subordinates, disrupting their work rhythm [40] and leading to more frequent experiences of negative emotions such as tension and anxiety [19] and more emotional exhaustion [34].

Finally, a leader’s bottom-line mentality is likely to destroy subordinates’ psychological security at work. A leader with a high bottom line mentality will send a signal to employees that “winners are winners, losers are losers” and stimulate competition among employees. [18]. Under the guidance of such information, when subordinates fail at work, they will feel a great risk of being overtaken or even displaced [20]. As a result, subordinates who work under a leader with a high bottom-line mentality may feel that they are merely tools for the leader or the organization to achieve profits and thus experience greater instrumentality [20]. This process of work alienation can damage subordinates’ moral commitment to their work [21], increase job insecurity, and even lead to higher turnover intentions [20].

To sum up, we posit that, as an important influencing factor in the workplace, a leader with a bottom-line mentality leads to more depletion of physical energy, more emotional exhaustion, and more job insecurity among subordinates, thereby reducing their psychological availability. Thus, we propose the following hypothesis:

**Hypothesis** **1.**
*There is a negative relationship between leader bottom-line mentality and subordinate psychological availability.*


### 2.3. Psychological Availability and Learning from Work Failures

Social information processing theory suggests that an individual’s perception and interpretation of external signals will influence their subsequent behavioral responses [22]. Learning from work failures means that employees can identify potential causes of failure events and proactively learn from them to enhance their knowledge and skills so that they can be successful the next time they encounter a similar incident [2,34,35]. Existing research has shown that positive cognitions, affective states, and motivations in their work can better drive individuals to learn from failure [2]. Psychological availability has been shown to induce a number of positive workplace outcomes (e.g., organizational citizenship behavior; work engagement) [25], as it can bring numerous positive resources to individuals [41]. In the context of work failure, we argue that these workplace resources will make employees more willing to learn from work failure.

First, high levels of psychological availability can inject more physical energy into individuals [42]. Abundant work energy can provide individuals with more vitality [43], which helps them maintain a positive and healthy physical state at work. When faced with failure events, individuals with work energy can have more sufficient physical resources to invest in thinking and learning from work failures [44]. Second, individuals with high levels of psychological availability tend to have sufficient positive cognitive patterns toward failure [45]. For example, they may have flexible cognition toward their work and keep an open mind toward failure events, which can help them focus on the failure event after encountering a failure and quickly engage in coping with the failure [46]. Third, individuals with higher psychological availability usually have rich emotional resources, and they can maintain an optimistic and positive attitude toward failure, which helps them maintain a healthy mentality and engage in positive behaviors that are beneficial to learning from work failures [30]. In addition, they have strong resilience and can adjust negative emotions in time when facing setbacks and challenges, which helps them to reflect on failures more quickly [47]. In contrast, when individuals have low levels of psychological availability, they have fewer physical, cognitive, and emotional resources and are unable to focus their limited resources on their work, which robs them of confidence that they can adequately cope with failure [48]. For these reasons, we suggest that a high level of psychological availability is a prerequisite for promoting learning from work failures. Therefore, we propose the following hypothesis:

**Hypothesis** **2.**
*Psychological availability has a positive relationship with learning from work failures.*


Based on the social information processing theory, individuals’ attitudes and behaviors are not only determined by their own needs and goals but are also strongly influenced by the social environment around them [22]. Combining Hypotheses 1 and 2, we argue that the bottom-line mentality of a leader will send a signal to focus only on bottom-line goals, which will have a negative effect on subordinates from the aspects of physical energy, emotions, and job security, which are crucial to stimulating subordinates’ learning from work failure. Therefore, we propose the following hypothesis:

**Hypothesis** **3.**
*Leader bottom-line mentality has a negative indirect relationship with subordinates learning from work failures via the mediating role of subordinate psychological availability.*


### 2.4. The Moderating Role of Subordinate Self-Compassion

According to the social information processing theory, individuals will adjust their cognitions and attitudes to fit their environment [22]. Leaders’ bottom-line mentality is a typical contextual factor that sends signals to subordinates that they are overly focused on bottom-line goals and company profits, further negatively affecting their attitudes and cognitions. Self-compassion refers to how individuals relate to themselves in cases of perceived failure, inadequacy, or personal suffering [27], which is closely related to the context of failure. We argue that self-compassion, as a productive way of dealing with distressing thoughts and emotions, can moderate the relationship between leader bottom-line mentality and subordinate psychological availability.

First, high self-compassion represents a high level of psychological resilience, which can help individuals resist the effects of negative emotions caused by leader bottom-line mentality. Leader bottom-line mentality can reduce subordinates’ work motivation by generating more negative emotions. Individuals with high levels of self-compassion approach and respond to negative emotions in a positive way, rather than avoiding and engaging in them [27]. Therefore, self-compassion can reduce an individual’s negative mindset and increase positive psychological experiences such as happiness, hope, gratitude, curiosity, and vitality [49,50]. In addition, self-compassion is closely related to an individual’s resilience [51], which refers to an individual’s ability to bounce back quickly from negative emotions [52]. Therefore, individuals with high levels of self-compassion are not overwhelmed and isolated by negative emotions [28] and are better able to cope with the negative effects of negative emotions.

Second, individuals with high self-compassion tend to focus more on the importance of the outcome for themselves rather than the outcome itself, thereby avoiding resource consumption. Self-compassion can promote an individual’s broaden-and-build process by enhancing a positive psychological state [53], which can focus an individual’s attention on doing the right thing. Individuals with high levels of self-compassion are more likely to focus on whether the goal is meaningful to them rather than blindly pursuing the success of the goal [54], which can avoid the loss of resources caused by the failure of the overly focused outcome. In addition, self-compassion is also an important source of motivation that can provide resources to individuals through warmth, encouragement, and constructive feedback to help them achieve their goals [27].

Third, individuals with high levels of self-compassion focus not only on themselves but also on others [27]. Self-compassion is not selfish or self-centered, and being kind and caring to oneself does not mean not caring about others. Instead, self-compassion can strengthen connections with others [55]. Individuals with high levels of self-compassion are more likely to provide social support and interpersonal trust to others and are more likely to offer help and warmth to others [56]. Therefore, individuals with high levels of self-compassion may not only have positive effects on themselves but also provide emotional resources to others by caring for and interacting with others [27]. This positive interaction process mitigates the decrease in their psychological availability caused by the leader’s bottom-line mentality.

In summary, individuals with high levels of self-compassion are able to respond to negative emotions in a positive way, focusing more on the importance of the outcome for themselves rather than blindly pursuing success. They can also provide social support and emotional resources to other members of the team, effectively mitigating the decline in psychological availability caused by the leader’s bottom-line mentality. Therefore, we propose the following hypothesis:

**Hypothesis** **4.**
*Subordinate self-compassion moderates the negative relationship between leader bottom-line mentality and subordinate psychological availability. Specifically, this negative relationship will be weakened when the level of subordinate self-compassion is high.*


### 2.5. An Integrated Model

Social information processing theory suggests that individuals can adjust their own cognition and attitude towards their environment, thereby affecting their interpretation of external signals and thus affecting subsequent behaviors. Based on the above hypotheses, we argue that the negative indirect relationship between leader bottom-line mentality and subordinates learning from work failures is moderated by subordinate self-compassion. Specifically, a subordinate high in self-compassion can attenuate the negative effect of leader bottom-line mentality on subordinates learning from work failures via the mediating role of subordinate psychological availability. Therefore, we predict the following hypothesis:

**Hypothesis** **5.**
*Subordinate self-compassion moderates the indirect effect of leader bottom-line mentality on subordinates learning from work failures via the mediating role of subordinate psychological availability. Specifically, this mediated relationship will be weakened when the level of subordinate self-compassion is high.*


## 3. Method

### 3.1. Samples and Procedure

Failure has different possibilities in different organizational and industrial contexts. In previous studies on learning from failure, researchers mostly chose industries with typical or high failure possibilities for research. Specifically, because high-tech industries are characterized by high risks and high returns, and a large proportion of R&D personnel, failure is more common in such types of industry. In addition, organizational learning is an important path to promote the long-term development of high-tech industry. Based on this logic, some scholars have used such enterprises as research samples in the context of failure research [11,15]. Additionally, the effectiveness of leader characteristics depends on the organizational context [57]. Because of the fast development pace, high proportion of small- and medium-sized enterprises, and intense industry competition in China’s high-tech enterprises, being able to maintain a high level of profits in the market is an important prerequisite for the success of firms in these industries, which may enable managers to have a higher level of bottom-line mentality to ensure the achievement of firms’ goals. Therefore, the high-tech industry also provides an organizational context for some researchers to understand the antecedents and consequences of leader bottom-line mentality [37,58].

Out of consideration for the above, we randomly selected a sample of employees working in high-tech R&D units in Beijing and Shanghai, China, and obtained more samples through snowball sampling. These employees are all from frontline R&D positions in industries such as the Internet, smartphone development, and intelligent devices. All subjects gave their informed consent for inclusion before they participated in this study. This study was conducted in accordance with the Declaration of Helsinki, and the protocol was approved by the Ethics Committee of the School of Government, Beijing Normal University (ZGY064; 1 December 2022).

To avoid common method bias, we conducted a three-wave time-lagged survey. At Time 1, we collected participants’ judgments of their immediate supervisors’ bottom-line mentality and their self-reported self-compassion. Two weeks later, at Time 2, we collected participants’ self-reported psychological availability. Another two weeks later, at Time 3, we collected participants’ self-reported learning from work failures and their demographic information.

At Time 1, we collected responses from 307 participants; at Time 2, we collected responses from 278 participants, with a response rate of 90.6%; at Time 3, we collected responses from 269 participants, with a response rate of 87.6%. Finally, we matched the samples that responded in all three collections, for a total of 245 responses, for a response rate of 79.8%. Among these participants, the average age was 33.0 years, and the average length of service was 8.86 years. In terms of gender, 62.8% were male and 37.2% were female. In terms of educational level, vocational/technical school accounted for 0.4%, high school accounted for 2.9%, junior college accounted for 7.8%, bachelor’s degree accounted for 71.8%, master’s degree accounted for 16.7%, and doctoral degree accounted for 0.4%.

### 3.2. Measures

The scales we used were sourced from mature scales in international top management journals which have been widely adopted by numerous scholars and have demonstrated good reliability and validity. To ensure that the meaning of the scales was not altered, we employed Brislin’s [59] back-translation method during the measurement process. Two professors in management research with a strong foundation in English translated the scales from English to Chinese, and then another English-proficient professor translated them back from Chinese to English. The meaning of the scales remained unchanged. All scales were measured using a 5-point Likert scale (from “1 = totally disagree” to “5 = totally agree”), and all items of the scales are listed in Appendix A.

#### 3.2.1. Leader Bottom-Line Mentality

Leader bottom-line mentality was measured by the 4-item scale developed by Greenbaum et al. [18]. A sample item is “My immediate leader is solely concerned with meeting the bottom line”. The Cronbach’s alpha of this scale is 0.82. The composite reliability (CR) was 0.80, and the average variance extracted (AVE) was 0.50, which means an acceptable level of reliability and validity was reached.

#### 3.2.2. Psychological Availability

Psychological availability was measured by the 7-item scale developed by Byrne et al. [60]. A sample item is “I am able to do the thinking that is necessary to do my work”. The Cronbach’s alpha of this scale is 0.89. The CR was 0.88, and the AVE was 0.51, which means an acceptable level of reliability and validity was reached.

#### 3.2.3. Learning from Work Failures

Learning from work failures was measured by the 5-item scale developed by Carmeli [35]. A sample item is “I ask questions such as ‘is there a better way to produce the product or provide the service?”. The Cronbach’s alpha of this scale is 0.86. The CR was 0.85, and the AVE was 0.52, which means an acceptable level of reliability and validity was reached.

#### 3.2.4. Self-Compassion

Self-compassion was measured by the 12-item scale developed by Raes et al. [61]. A sample item is “I try to see my failings as part of the human condition.” The Cronbach’s alpha of this scale is 0.93. The CR was 0.92, and the AVE was 0.50, which means an acceptable level of reliability and validity was reached.

#### 3.2.5. Control Variables

We took gender (1 = men, 2 = women), age, organizational tenure, education level (1 = vocational/technical school, 2 = high school, 3 = junior college, 4 = bachelor’s degree, 5 = master’s degree, 6 = doctoral degree) as control variables, as previous studies have demonstrated that these variables may correlate with learning from work failures.

## 4. Results

### 4.1. Descriptive Statistics and Correlations

Means, standardized deviations, and correlations among study variables are shown in Table 1. The results showed that leader bottom-line mentality has a negative relationship with psychological availability (r = −0.45, *p* < 0.01) and learning from work failures (r = −0.42, *p* < 0.01). Furthermore, psychological availability has a positive relationship with learning from work failures (r = 0.64, *p* < 0.01). The results provided preliminary evidence for Hypothesis 1, 2, and 3.

### 4.2. Confirmatory Factor Analysis

We conducted a series of confirmation factor analyses (CFAs) to test the discriminant validity of the study variables. Compared with the index data of the competitive models, the fitting degree of the four-factor model is the most ideal (χ^2^ = 896.25, df = 410, χ^2^/df = 2.19, CFI (comparative fit index) = 0.92, TLI (Tucker–Lewis index) = 0.91, RMSEA (root-mean-square error of approximation) = 0.06, SRMR (standardized root-mean-square residual) = 0.06), which shows that the four study variables (i.e., leader bottom-line mentality, psychological availability, self-compassion, and learning from work failures) in this study have good discriminant validity. The results of CFAs are shown in Appendix B.

### 4.3. Testing Common Method Bias

As all data were reported by employees, the salience of common method bias may still exist. The results of Harman’s one-factor test [62] indicated that the first factor accounts for 33% of total variance, which is lower than the 50% standard. Therefore, the issue of common method bias is not significant in our study.

### 4.4. Hypothesis Tests

#### 4.4.1. Testing Mediating Effect

To further test the relationship between the studied variables, following Baron and Kenny’s suggestion [63], we used the causal steps approach to conduct a series of hierarchical regression analyses using SPSS 26.0 (as shown in Table 2). The causal steps approach is considered the most commonly used method by numerous management, psychology, and sociology scholars to test mediation and moderation models [64]. The results reveal that there is a negative relationship between leader bottom-line mentality and psychological availability (B = −0.21, SE = 0.03, *p* < 0.001, see Model 2), and psychological availability has a positive relationship with learning from work failures (B = 0.56, SE = 0.05, *p* < 0.001, see Model 6); thus, Hypotheses 1 and 2 are supported. 

Additionally, Hypothesis 3 posited a mediating effect of psychological availability in the relationship between leader bottom-line mentality and learning from work failures. According to MacKinnon et al. [65], on the basis of the causal steps approach, it is necessary to further confirm the statistical significance of the mediating effect through the products of coefficient tests. We used the macro program PROCESS developed by Hayes [66] to conduct a Bootstrap test of the indirect effect of leader bottom-line mentality on learning from work failures. In total, 1000 Bootstrap samples were randomly selected by repeated random sampling based on the original data (*n* = 245), and 95% confidence intervals were obtained by estimating the mediating effect. The results show that the estimated value of the indirect effect is −0.14, and the 95% CI is [−0.285, −0.046], excluding 0. Therefore, Hypothesis 3 is also supported.

#### 4.4.2. Testing Moderating Effect

As shown in Model 3, we conducted an interaction term (leader bottom-line mentality × psychological availability) to test the moderating role of self-compassion in the relationship between leader bottom-line mentality and psychological availability. The results indicate that the interaction term has a significant and positive relationship with psychological availability (B = 0.49, SE = 0.25, *p* < 0.05), which supports the existence of the moderating effect. To clearly demonstrate the direction of the moderating effect, we plotted a moderation effect paradigm (as shown in Figure 2) based on the influence of the independent variable (i.e., leader bottom-line mentality) on the dependent variable (i.e., psychological availability) when the moderating variable (i.e., self-compassion) was grouped into high and low levels. Specifically, the negative relationship between leader bottom-line mentality and psychological availability is weaker (b_high_ = −0.20, SE_high_ = 0.03, *p*_high_ < 0.001; b_low_ = −0.30, SE_low_ = 0.04, *p*_low_ < 0.001) when self-compassion is high (mean + 1SD) than when self-compassion is low (mean—1SD), indicating that self-compassion can alleviate the negative relationship between leader bottom-line mentality and psychological availability. Therefore, Hypothesis 4 is supported.

Furthermore, Hypothesis 5 posited that self-compassion can moderate the negative and indirect effect of leader bottom-line mentality on subordinates learning from work failures through subordinate psychological availability. A test of differences in coefficients (index of moderated mediating effect) should be used to test the moderating effect of the moderator on the mediating process [65]. According to the results of the PROCESS macro program, the moderated mediation effect was significant with an index value of 0.08 and a 95% CI of [0.006, 0.279]. Therefore, Hypothesis 5 is supported.

#### 4.4.3. Robustness Check

Combining multiple methods to test research hypotheses can enhance the robustness of research findings. Referring to the suggestions of Wood et al. [64], we further used the structural equation modeling (SEM) method to conduct a path analysis on each hypothesis of the research model. According to the analysis results of Mplus 8.0 (as shown in Figure 3), the significance of each path was supported. Furthermore, the estimated value of the indirect effect is −0.14, and the 95% CI is [−0.267, −0.047]. The moderated mediation effect is 0.08, and the 95% CI is [0.004, 0.283]. Therefore, the results of the path analysis indicate that our findings are sufficiently robust.

## 5. Discussion

Although numerous studies have paid attention to the antecedents that affect individual learning from work failures, scarce research focuses on the crucial role of the leader in affecting subordinates learning from work failures. Unlike the limited existing research that focuses on positive factors of leaders, the current study focuses on leader-related factors that may hinder subordinates learning from work failures. Based on social information processing theory, we constructed a moderated mediation model. The results showed that leader bottom-line mentality will impede subordinates learning from work failures by reducing subordinate psychological availability. Furthermore, we identified the boundary role of subordinate self-compassion in this mediated relationship. Specifically, the negative relationship between leader bottom-line mentality and subordinates learning from work failures via the mediating role of subordinate psychological availability is weaker when the level of subordinate self-compassion is higher. Our study provides some implications for the theoretical literature and managerial practice.

### 5.1. Theoretical Implications

The present study has several theoretical implications for the current related research. First, we further enrich the literature on subordinates learning from work failures and leader bottom-line mentality. Dahlin et al. [2] posited that leader-related factors are important motivational antecedents that affect employees learning from failure. Therefore, in recent years, some scholars have paid attention to how leadership style or leader behaviors can stimulate employees’ learning from work failures [13,15,67]. However, these studies are not comprehensive, ignoring the potential dark side of leaders, and they lack the exploration of leader-related factors that may hinder employees from learning from work failures. Based on social information processing theory, we link the literature on learning from failure to leader bottom-line mentality and examine the detrimental effects of leader bottom-line mentality on subordinates learning from work failures. Indeed, the multi-stream literature shows that leader bottom-line mentality is closely related to failure avoidance [20]. Our research reveals how leader bottom-line mentality to avoid financial losses and reduced performance sends negative signals to subordinates, which in turn prevents subordinates from learning from work failures. Therefore, we not only extended the discussion on the antecedents of learning from failure but also extended the leader bottom-line mentality literature to failure research, thus expanding the application scenarios of the leader bottom-line mentality literature.

Second, by examining the mediating mechanisms of psychological availability, we identify and enrich the key mediators linking leader bottom-line mentality and subordinates learning from work failures. Social information processing theory has been used by a large number of scholars to explain the process by which leader bottom-line mentality affects employee outcomes [20], and leader-related factors affect subordinates learning from failure [13,15]; these studies explain how leader-related factors affect subordinate motivation (e.g., intrinsic motivation) [13] and behavioral responses (e.g., perspective taking) [15], causing subordinates to learn from work failures. However, these studies ignore the triggering process of psychological states that are relatively close to leader-related factors. Our research focuses on how negative leader behaviors hinder subordinates’ learning from work failures by affecting their psychological state (i.e., psychological availability), further enriching the micro-psychological mechanism process of leader-related factors affecting subordinates’ learning from work failures and applying social information processing theory to failure management research.

Third, we identify a key individual difference factor (i.e., self-compassion) that causes a leader’s bottom-line mentality to influence subordinates’ processes of learning from work failures. This research echoes Shepherd et al. [35] who believe that individual ability and willingness will have an impact on learning from failure. As a key self-regulatory strategy, self-compassion is acknowledged to have an important buffering effect on individuals (e.g., pain, failure, and grief) caused by negative situations [68]. Although some scholars have pointed out that self-compassion can reduce negative effects such as emotional exhaustion and cynicism on individuals [69] and improve positive effects such as job performance and job satisfaction [70,71], organizational scholars have not paid enough attention to the key regulatory role of self-compassion in individuals in the context of failure. Our study applied the self-compassion literature to failure research, further tested the buffering effect of self-compassion on individuals experiencing negative psychological states, and further expanded the application scenarios of the self-compassion literature.

### 5.2. Practical Implications

On the one hand, leader bottom-line mentality may affect subordinates learning from work failures, and leaders should avoid paying excessive attention to promoting performance. Learning from work failures has been shown to have a positive impact on employees’ ability to proactively identify problems, learn knowledge, and master skills [4,33]. Therefore, as leaders of organizations and teams, it is important to avoid a bottom-line mentality and not signal to subordinates that they are overly concerned with the bottom line and profits. More attention should be paid to caring for and supporting employees during the work process, helping them to effectively deal with and learn from failures, and creating a harmonious and good working atmosphere to promote the long-term development of the organization.

On the other hand, employees’ self-compassion can alleviate the decrease in psychological availability caused by leaders’ bottom-line mentality and is an effective strategy for employees to cope with failure and bottom-line mentality. In their daily work, employees should focus on the importance of the goals they are pursuing for themselves, rather than just pursuing the success of the goals and achieving bottom-line goals. Reducing the reliance on results can avoid the waste of resources caused by an over-focus on results and focus limited resources on more meaningful things. At the same time, employees should also pay attention to cultivating the ability of self-compassion, actively provide social support and interpersonal trust for themselves and others, offer help to others, and create a good atmosphere of mutual help.

### 5.3. Limitations and Future Research Directions

Although the present study makes theoretical and practical contributions, several limitations must be acknowledged. First, we used a time-lagged design to mitigate common method bias (CMB), as suggested by Podsakoff et al. [72]. However, this does not allow for any causal inferences to be made about the research hypotheses. Future research could combine causal explanations for the research findings using methods such as experimental design and the cross-lagged method to further enhance the validity of the study.

Second, another potential limitation is the generalizability of our results. While we tested our research model through a three-wave survey conducted in high-tech firms, work failures are common in organizational management, but learning from work failures is important for any type of organization [4]. We hope to replicate the conclusions of our study in a variety of organizational types in future research. Similarly, our study only tested the hypotheses with the sample from China, which does not provide empirical evidence on the generalizability of the research findings in different cultural and national contexts. Future research needs to examine the boundary conditions (e.g., collectivist culture versus individualist culture) that more national culture or value factors may play in our research model to enhance the generalization of our research. 

Finally, although we believe that a leader’s bottom-line thinking can prevent subordinates from learning from work failures by reducing their psychological availability, existing research provides further potential alternative explanations for our study. For example, a leader’s bottom-line thinking may strengthen subordinates’ sense of goals [73], which may make them more invested in improving their job performance [74]. In addition, some studies believe that leader bottom-line mentality will lead subordinates to adopt a mental preoccupation with work, which may make subordinates focus on proactively learning from work tasks to further improve their work efficiency and work quality [17]. Therefore, these studies provide some empirical evidence that leader bottom-line mentality may also induce positive consequences. Future research can try to unravel when and how leader bottom-line mentality may have a positive effect on subordinates learning from work failures.

## Figures and Tables

**Figure 1 behavsci-14-00226-f001:**
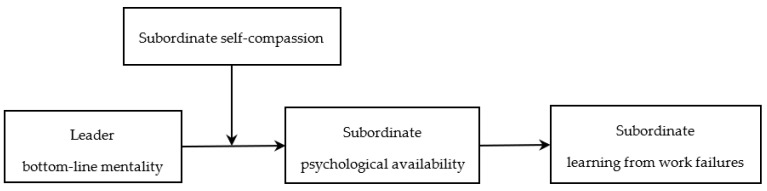
Theoretical framework.

**Figure 2 behavsci-14-00226-f002:**
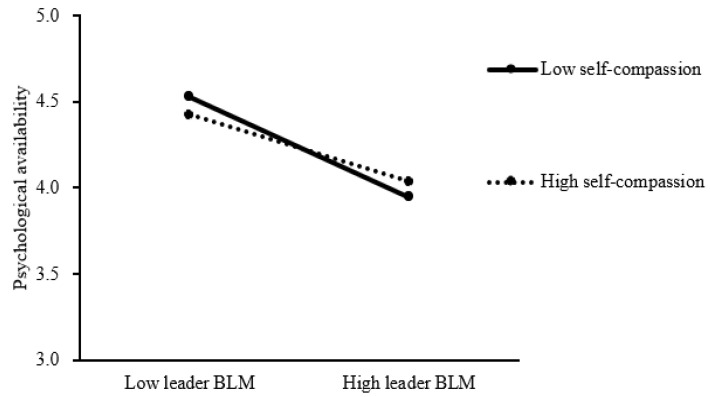
The moderating role of subordinate self-compassion. Note: leader BLM = leader bottom-line mentality.

**Figure 3 behavsci-14-00226-f003:**
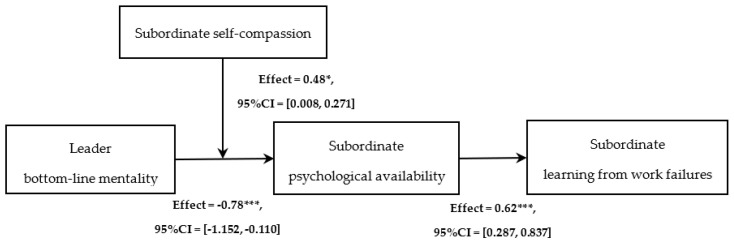
Path analysis results. Note: * *p* < 0.05, *** *p* < 0.001.

**Table 1 behavsci-14-00226-t001:** Means, standardized deviations, and correlations.

Variables	Means	SD	1	2	3	4	5	6	7	8
1 Gender	1.63	0.48								
2 Age	33.01	6.96	−0.19 **							
3 Education	4.03	0.64	−0.10	−0.26 **						
4 Organizational tenure	8.86	6.36	−0.20 **	0.93 **	−0.32 **					
5 Leader bottom-line mentality	2.17	0.99	−0.15 *	0.08	−0.19 **	0.09	(0.82)			
6 Psychological availability	4.25	0.46	0.04	0.15 *	0.09	0.11	−0.45 **	(0.89)		
7 Self-compassion	3.25	0.35	−0.01	0.01	−0.09	−0.02	0.37 **	−0.13 *	(0.93)	
8 Learning from work failures	4.28	0.46	0.01	0.04	0.24 **	−0.02	−0.42 **	0.64 **	−0.10	(0.86)

Note: sample size = 245; * *p* < 0.05, ** *p* < 0.01; Cronbach’s alpha for each study variable is in parentheses on the diagonal.

**Table 2 behavsci-14-00226-t002:** Hierarchical regression analysis results.

Variables	Psychological Availability						Learning from Work Failures					
	Model 1		Model 2		Model 3		Model 4		Model 5		Model 6	
	*B*	*SE*	*B*	*SE*	*B*	*SE*	*B*	*SE*	*B*	*SE*	*B*	*SE*
Constant	3.09 ***	0.36	3.92 ***	0.33	5.92 ***	1.17	2.87 ***	0.35	3.59 ***	0.34	1.42 ***	0.35
*Control variables*												
Gender	0.09	0.06	0.01	0.06	0.01	0.06	0.05	0.06	−0.02	0.06	−0.03	0.05
Age	0.02	0.01	0.02	0.01	0.02	0.01	0.02	0.01	0.02	0.01	0.01	0.01
Education	0.10 *	0.05	0.04 *	0.04	0.03	0.04	0.19 *	0.05	0.13 *	0.04	0.11	0.04
Organizational tenure	−0.01	0.01	−0.01	0.01	−0.01	0.01	−0.02 ***	0.01	−0.02 **	0.01	−0.01	0.01
*Predictors*												
Leader bottom-line mentality			−0.21 ***	0.03	−0.66 **	0.22			−0.19 ***	0.03	−0.07 **	0.03
Self-compassion					−0.27	0.19						
Leader bottom-line mentality × Self-compassion					0.49 *	0.25						
Psychological availability											0.56 ***	0.05
*F*	3.24 *		15.58 ***		11.87 ***		5.30 ***		14.42 ***		34.49 ***	
*R* ^2^	0.05		0.24		0.26		0.08		0.23		0.47	
Δ*R*^2^	—		0.19		0.02		—		0.15		0.24	

Note: sample size = 245; * *p* < 0.05, ** *p* < 0.01, *** *p* < 0.001.

## Data Availability

The data sets generated during and/or analyzed during the current study are available from the corresponding author on reasonable request.

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
