# Peer review of "Exploring the Impact of Leader Bottom-Line Mentality on Subordinate Learning from Work Failures: A Social Information Processing Perspective"

_behavsci, 2024, doi:10.3390/bs14030226_

Round 1

Reviewer 1 Report

Comments and Suggestions for Authors

This is an interesting paper that discusses a highly relevant topic in organisational leadership. The paper is clearly structures and well written. 

A few amendments should be made:

1. Key terms need clearer definitions, for example self-compassion

2. It would be good to show the full 4-factor CFA model in an appendix

3. It is not clear why the authors chose to d a series of regression models to test their hypotheses, rather than use a structural equation model. This choice either needs clarifying or an SEM used instead. 

Reviewer 2 Report

Comments and Suggestions for Authors

This article examines the detrimental impact of leader bottom-line mentality on subordinate learning from work failures, highlighting the mediating role of subordinate psychological availability and the moderating effect of self-compassion, thus providing valuable insights for both theoretical understanding and practical implications in organizational settings.

The introduction of the article sets the stage for exploring an important and relatively understudied aspect of organizational behavior: how leaders' bottom-line mentality influences subordinates' learning from work failures. The authors effectively argue that learning from failures is crucial for individual and organizational development, and they introduce the concept of bottom-line mentality as a potential hindrance to this learning process. The integration of social information processing theory to frame their research questions and hypotheses adds theoretical depth to their study.

One strength of the introduction is its clear articulation of the research gap: while previous literature has focused on factors that facilitate learning from failures, there's a lack of attention given to factors, specifically leader-related, that may impede this learning process. By addressing this gap, the authors contribute to a more comprehensive understanding of organizational learning dynamics.

Furthermore, the introduction provides a concise overview of the theoretical framework and key constructs under investigation, such as psychological availability and self-compassion. This clarity aids readers in understanding the conceptual underpinnings of the study and how these constructs interact within the proposed model.

However, there are a few areas where the introduction could be strengthened:

The article mentions that the study was conducted in high-tech companies in China, but it lacks further contextualization regarding why this setting was chosen and how the cultural or industry-specific factors might influence the phenomena under investigation. Providing this context would enhance the generalizability and relevance of the findings.

Whereas the literature review is generally well-structured and informative, there are some weaknesses that could be addressed to enhance its quality:

Given that the study was conducted in high-tech companies in China, a more in-depth discussion of how cultural values or organizational contexts might shape the relationships between leader bottom-line mentality, psychological availability, and learning from failures would enrich the analysis.

- While the review integrates social information processing theory, there is limited discussion on how this theory specifically informs the study's hypotheses beyond surface-level explanations. Providing more detailed insights into the theoretical mechanisms and pathways through which leader bottom-line mentality affects subordinate behavior and outcomes would enhance the theoretical robustness of the study.

- There are instances of repetition and redundancy in the literature review, particularly in the sections discussing the impact of leader bottom-line mentality on subordinates. Streamlining the presentation of key concepts and findings would improve the clarity and coherence of the review.

Addressing these weaknesses would strengthen the literature review by providing a more nuanced and comprehensive analysis of the existing research landscape, enhancing the theoretical and methodological foundations of the study, and ultimately contributing to the credibility and validity of the study's findings.

Overall, the discussion section of the study provides valuable insights into the theoretical and practical implications of the research findings. However, there are areas where the discussion could be further strengthened:

While the discussion highlights the theoretical implications of the research findings, there is a tendency to focus predominantly on the positive aspects. It would be beneficial to also consider potential limitations or challenges associated with the proposed theoretical frameworks or findings. Acknowledging potential counterarguments or alternative interpretations would provide a more balanced perspective.

The discussion briefly mentions the use of a time-lagged design to mitigate common method bias but does not delve into other methodological limitations of the study. A more thorough discussion of potential methodological weaknesses, such as sample characteristics, measurement validity, or alternative research designs, would enhance the transparency and rigor of the research.

Although the study acknowledges the importance of replicating the findings in different organizational contexts, more discussion on the generalizability of the results to diverse settings would be beneficial. Exploring potential boundary conditions or contextual factors that might influence the observed relationships could provide deeper insights into the external validity of the findings.
